# Endothelial IGF-1 receptor mediates crosstalk with the gut wall to regulate microbiota in obesity

Natalie J Haywood[1] (ID), Cheukyau Luk[1], Katherine I Bridge[1], Michael Drozd[1], Natallia Makava[1],
Anna Skromna[1], Amanda Maccannell[1], Claire H Ozber[1], Nele Warmke[1], Chloe G Wilkinson[1],
Nicole T Watt[1] (ID), Joanna Koch-Paszkowski[1], Irvin Teh[1], Jordan H Boyle[2], Sean Smart[3],
Jurgen E Schneider[1], Nadira Y Yuldasheva[1], Lee D Roberts[1], David J Beech[1], Piruthivi Sukumar[1],
Stephen B Wheatcroft[1], Richard M Cubbon[1] & Mark T Kearney[1],* (ID)

## Abstract

**Changes in composition of the intestinal microbiota are linked to the development of obesity and can lead to endothelial cell (EC) dysfunction. It is unknown whether EC can directly influence the microbiota. Insulin-like growth factor-1 (IGF-1) and its receptor (IGF-1R) are critical for coupling nutritional status and cellular growth; IGF-1R is expressed in multiple cell types including EC. The role of ECIGF-1R in the response to nutritional obesity is unexplored. To examine this, we use gene-modified mice with EC-specific overexpression of human IGF-1R (hIGFREO) and their wild-type littermates. After high-fat feeding, hIGFREO weigh less, have reduced adiposity and have improved glucose tolerance. hIGFREO show an altered gene expression and altered microbial diversity in the gut, including a relative increase in the beneficial genus *Akkermansia*. The depletion of gut microbiota with broad-spectrum antibiotics induces a loss of the favourable metabolic differences seen in hIGFREO mice. We show that IGF-1R facilitates crosstalk between the EC and the gut wall; this crosstalk protects against diet-induced obesity, as a result of an altered gut microbiota.**

**Keywords** endothelium; IGF-1R; microbiota; obesity
**Subject Categories** Metabolism; Microbiology, Virology & Host Pathogen Interaction; Signal Transduction
See also: **Z Bouman Chen & N Kaur Malhi** (May 2021)

## Introduction

In the intestine are trillions of microorganisms which are collectively described as the gut microbiota. The traditional dogma that the gut microbiota is pathogenic has evolved with an appreciation of its important role in the maintenance of human health (Lynch & Pedersen, 2016). Recent studies indicate that the gut microbiota is important in the metabolic response to changes in dietary composition (Backhed *et al*, 2004; Turnbaugh *et al*, 2006; Vrieze *et al*, 2012). Obesity secondary to excess calorie intake is a major risk factor for the development of a range of common disorders of human health including the following: type 2 diabetes (Guariguata *et al*, 2013), fatty liver (Yki-Järvinen, 2014) and a number of cancers (Gallagher & Leroith, 2015). While our understanding of the mechanisms underlying the development and complications of obesity remains incomplete, a role for adverse remodelling of the gut microbiota has recently emerged as an important factor in the unfavourable effects of the disorder in a range of tissues and organs (Backhed *et al*, 2004; Turnbaugh *et al*, 2006; Khan *et al*, 2016; Patterson *et al*, 2016; Castaner *et al*, 2018) including the vascular endothelium (Koren *et al*, 2011; Karlsson *et al*, 2012; Catry *et al*, 2018; Leslie & Annex, 2018; Amedei & Morbidelli, 2019). The endothelium, previously thought to be an inert monolayer, has emerged as a complex paracrine/autocrine organ, important in the regulation of a range of homeostatic processes (Lee *et al*, 2007; Ding *et al*, 2010; Kivelä *et al*, 2019; Tang *et al*, 2020). It is currently unknown whether the endothelium can influence the composition of the intestinal microbiota.

The insulin-like growth factors (IGF-I and IGF-II) are evolutionally conserved peptide hormones that couple nutrient intake to cellular growth (Jones & Clemmons, 1995). The effects of IGF-I are predominantly mediated by the activation of its plasma membrane receptor—IGF-1R (Adams *et al*, 2000). During calorie excess, the expression of IGF-1R changes in a range of tissues, including the endothelium, where we have shown it to decline (Mughal *et al*, 2019). The IGF-1R has also been shown to modulate the intestinal barrier (Dong *et al*, 2014), and conversely, the microbiome has been shown to modulate IGF-1R signalling in muscle (Schieber *et al*, 2015) and bone formation (Yan *et al*, 2016). Therefore, to explore the effects of endothelial IGF-1R on metabolic responses to obesity and the microbiome, we fed mice with endothelial cell

1   Faculty of Medicine and Health, Leeds Institute of Cardiovascular and Metabolic Medicine, University of Leeds, Leeds, UK
2   Faculty of Engineering, School of Mechanical Engineering, University of Leeds, Leeds, UK
3   Department of Oncology, University of Oxford, Oxford, UK
    *Corresponding author. Tel: +44 113 343 8834; E-mail: m.t.kearney@leeds.ac.uk

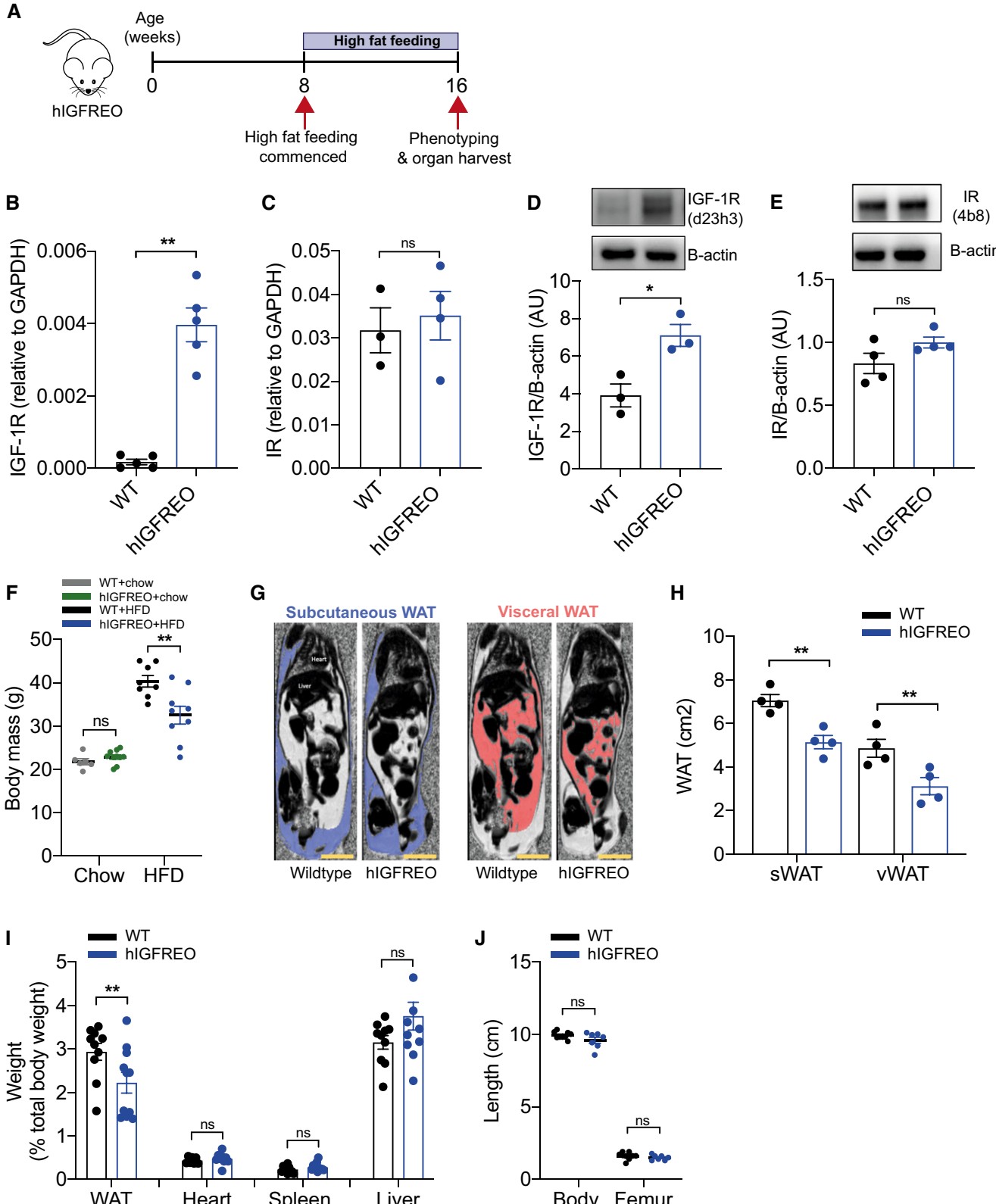

**Figure 1.**

◄

**Figure 1.  Endothelial IGF-1R overexpression prevents high-fat diet (HFD)-induced weight gain.**

A       Schematic representation of feeding time course.
B, C    In primary endothelial cells isolated from human IGF-1 receptor endothelial overexpressing mice (hIGFREO) and wild-type littermates (WT), quantitative polymerase chain reaction (qPCR) shows that hIGFREO have increased expression of human IGF-1R but similar levels of murine insulin receptor (IR) gene expression as WT ($n$ = 3–5 mice per group).
D, E    In primary endothelial cells isolated from WT and hIGFREO, immunoblotting shows that hIGFREO have increased expression of IGF-1R but similar levels of IR protein expression ($n$ = 3–4 mice per group).
F       Chow-fed hIGFREO had similar body mass to WT; however, hIGFREO did not gain as much weight as WT after 8 weeks of HFD ($n$ = 6–10 mice per group).
G       Representative images of difference in fat and water distribution shown by magnetic resonance (MR) imaging in hIGFREO and WT. Scale bar = 1 cm.
H       Subcutaneous white adipose tissue (sWAT) and visceral white adipose tissue (vWAT) volumes were reduced in hIGFREO ($n$ = 4 per genotype).
I       hIGFREO had reduced white epididymal adipose depot weight compared with WT; there was no difference in heart, spleen or liver weight ($n$ = 7–11 mice per group).
J       hIGFREO had similar whole-body and femur length as WT ($n$ = 7–9 mice per group).

Data information: Data shown as mean ± SEM, individual mice are shown as data points, $P < 0.05$ taken as being statistically significant using Student's $t$-test and denoted as * (** denotes $P \leq 0.01$, ns denotes not significant).
Source data are available online for this figure.

overexpression of human IGF-1R (hIGFREO) (Imrie *et al*, 2012) an obesogenic high-fat high-calorie diet. Feeding hIGFREO an obesogenic diet revealed a hitherto unrecognised mode of communication between the endothelium and the gut wall leading to favourable remodelling of the gut microbiota which protects against the development of diet-induced obesity and its adverse metabolic sequelae.

## Results and Discussion

### Endothelial IGF-1R overexpression prevents high-fat diet-associated weight gain

To explore the role of IGF-1R in the endothelium under circumstances recapitulating diet-induced obesity, we fed hIGFREO and wild-type littermates (WT) a 60% high-fat diet (HFD) for 8 weeks (Fig 1A). Endothelial overexpression of hIGF-1R was confirmed using qPCR (Fig 1B); endothelial insulin receptor expression was similar in hIGFREO and WT (Fig 1C); this expression pattern was recapitulated at the protein level (Fig 1D and E). Protein markers of vascular function (eNOS and AKT) in the aorta were unchanged between the genotypes (Fig EV1A and B). On chow diet, hIGFREO had similar weight to WT, as we have previously reported (Imrie *et al*, 2012); however, on HFD, hIGFREO did not gain as much weight as WT mice (Fig 1F). MRI was used to assess whole-body adiposity; hIGFREO had significantly less subcutaneous and visceral adipose tissue compared with WT on HFD (Fig 1G and H). Wet organ weight confirmed that hIGFREO had smaller white epididymal adipose depots than WT on HFD, with no difference in heart, spleen or liver

weight (Fig 1I). The IGF-1R is known to be an important regulator of foetal and postnatal growth (Woods *et al*, 1996; Garcia *et al*, 2014; Fujimoto *et al*, 2015; Juanes *et al*, 2015), and hIGFREO and WT mice had similar body and femur length (Fig 1J), demonstrating that endothelial IGF-1R overexpression did not cause growth retardation.

### Overexpression of endothelial IGF-1R prevents obesity-associated glucose intolerance

Chow-fed hIGFREO had similar glucose tolerance as WT (Fig EV1C–E). However, when challenged by a HFD, hIGFREO had significantly lower fasting blood glucose compared with WT (Fig 2A) and were also protected from the glucose intolerance seen in WT (Fig 2B and C). hIGFREO on HFD were also more insulin sensitive as shown using the homeostatic model assessment of insulin resistance (HOMA-IR) analysis (Fig 2D), which was associated with an increase in the expression of AKT and phosphorylation of AKT at serine 437 in skeletal muscle of hIGFREO (Fig EV1F and G). hIGFREO and WT had similar fasting plasma concentrations of IGF-I and insulin (Fig 2E and F). HFD-fed hIGFREO handled olive oil gavage more effectively over a 3-hr period postgavage with a significantly smaller increment in plasma triglycerides than WT (Fig 2G and H).

### Endothelial IGF-1R overexpression does not lead to changes in activity, food intake or energy expenditure

To further probe the mechanisms underpinning the anti-obesity and anti-diabetic effect of endothelial IGF-1R, metabolic cages were used to perform measurement of multiple metabolic parameters.

**Figure 2.  Endothelial IGF-1R overexpression prevents high-fat diet (HFD)-induced glucose intolerance.**

A       Human IGF-1R endothelial overexpressing mice (hIGFREO) had significantly lower fasting blood glucose compared with wild-type littermates (WT) after HFD ($n$ = 5–7 mice per group).
B, C    hIGFREO had reduced glucose intolerance compared with WT (as measured by glucose tolerance test and area under the curve (AUC)) ($n$ = 5–7 mice per group).
D       hIGFREO had improved insulin sensitivity compared with WT as shown by lower HOMA-IR score ($n$ = 9–10 mice per group).
E, F    hIGFREO and WT had similar fasting plasma IGF-1 and insulin concentrations ($n$ = 6–12 mice per group).
G, H    Percentage change in plasma levels of triglycerides after an olive oil oral gavage was reduced over the 3-h period postgavage in hIGFREO compared with WT and shown as area under the curve ($n$ = 10–12 mice per group).

Data information: Data shown as mean ± SEM and individual mice are shown as data points, $P < 0.05$ taken as being statistically significant using Student's $t$-test and denoted as * (** denotes $P \leq 0.01$, ns denotes not significant).

▶

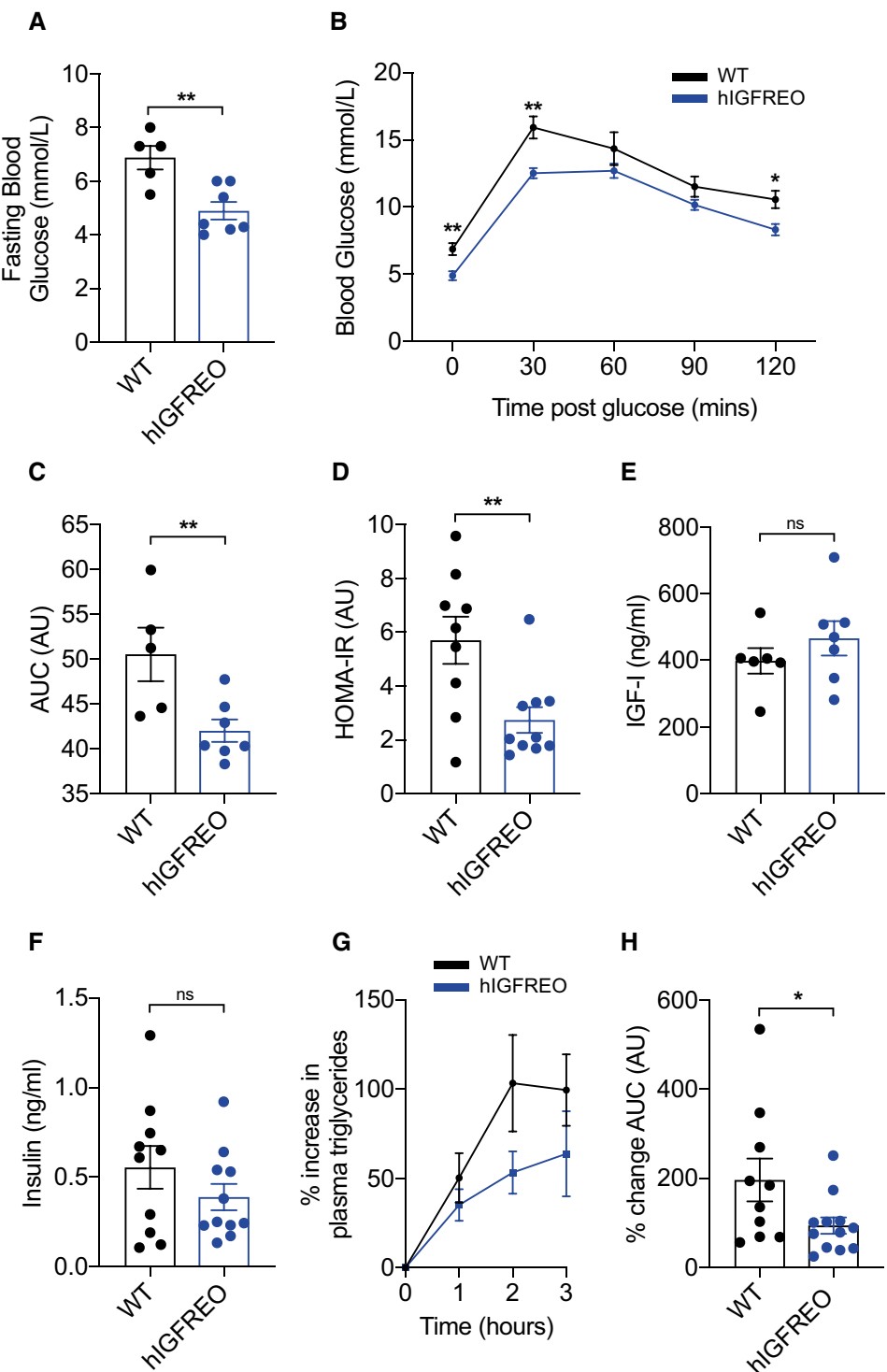

**Figure 2.**

hIGFREO on HFD showed no difference in activity levels (Fig 3A), food consumption (Fig 3B), oxygen consumption (Fig EV2A), carbon dioxide production (Fig EV2B), energy expenditure (Fig 3C) or respiratory exchange ratio (Fig EV2C) compared with WT on HFD. IGF-1R are thought to contribute to temperature homeostasis and may contribute to regulation of energy homeostasis during

calorie restriction (Cintron-colon *et al,* 2017). Going against this possibility adipose tissue expression of browning markers (Fig 3D) and body temperature (Fig 3E) were all unchanged in hIGFREO compared to WT. Plasma leptin and adiponectin were also no different (Fig EV2D and E). There was also no difference in adipose tissue remodelling, shown by similar adipocyte size (Fig EV3A–C),

adipose tissue vascularity (Fig EV3D and E) and adipose tissue inflammatory markers, in hIGFREO and WT on HFD (Fig EV3F–I). There was no difference in hepatic steatosis (Fig EV4A–H), pancreatic lipase or gene expression of cholesterol 7alpha-hydroxylase (Cyp7a) and ATP Binding Cassette Subfamily B Member 11(Abcb11) in liver when comparing hIGFREO to WT (Fig EV4I–K). There was no difference in small intestine length (Fig 3F), villi histology (Fig EV5A–C) or gut transit time (Fig 3G).

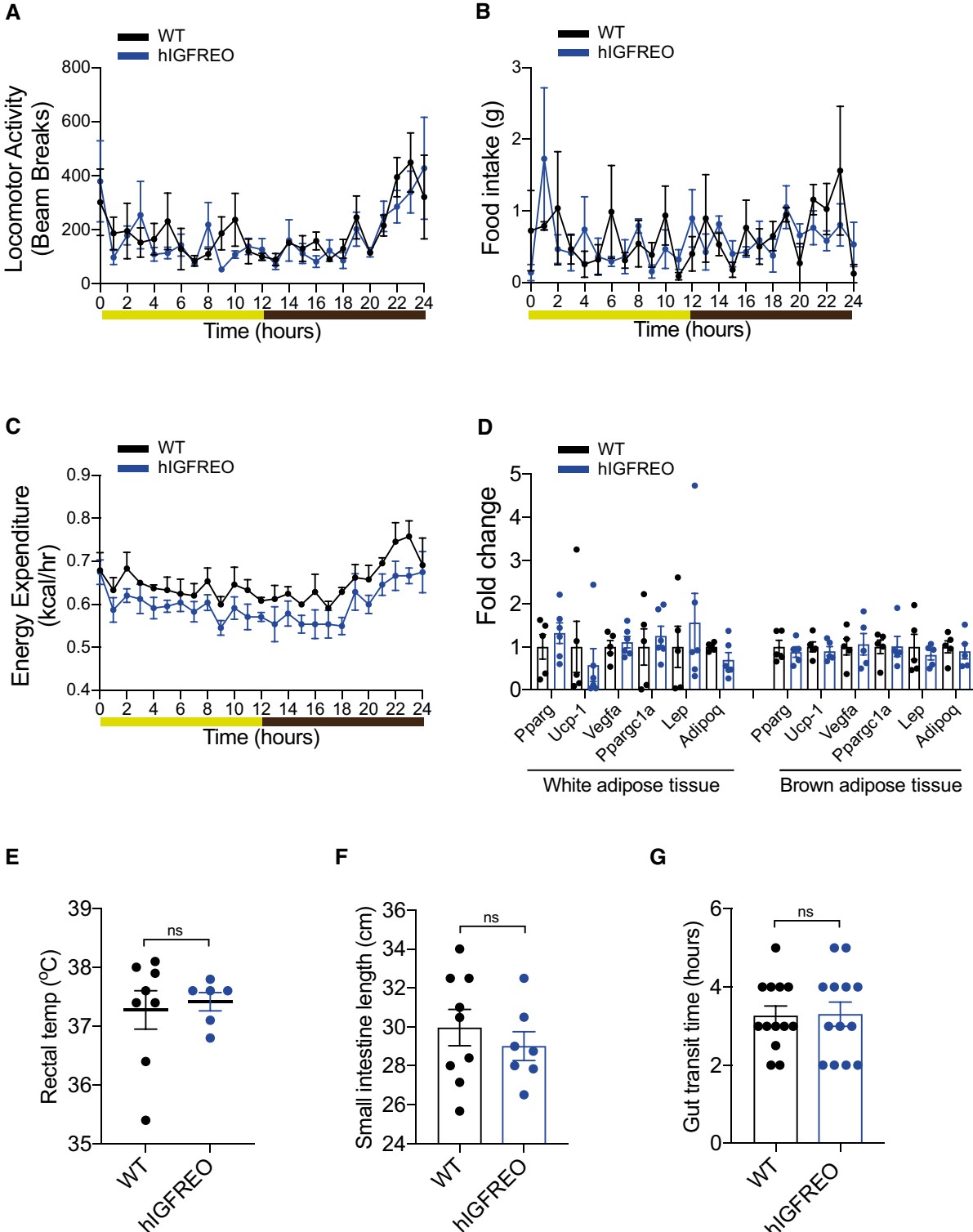

**Figure 3.**

Figure 3. Protection from high-fat diet (HFD)-induced weight gain in human IGF-1R endothelial overexpressing mice (hIGFREO) is not due to changes in activity, food intake, energy expenditure, adipose browning or gut transit time.

A–C   hIGFREO exhibit no difference in activity levels, food consumption or energy expenditure using indirect calorimeter assessment after HFD compared with wild-type littermates (WT) after HFD. (*n* = 4 per genotype).

D   Adipose expression of browning markers is also no different in white epididymal adipose tissue and brown adipose tissue compared with WT (*n* = 6 per genotype).

E   Core body temperature is no different in hIGFREO compared with WT (*n* = 6–8 mice per group).

F, G   Gut transit time is also unaltered in hIGFREO compared with WT as shown by no change in small intestine length (F) (*n* = 7–9 mice per group), or total gut transit time after a carmine red gavage (G) (*n* = 12–13 mice per group).

Data information: The light/dark cycle for graphs A–C is shown as follows: light in yellow and dark in brown. Data shown as mean ± SEM and individual mice are shown as data points. For indirect calorimetry, ANOVA testing was performed using mass as a co-variant (ANCOVA testing) using calrapp.org. ns denotes not significant.

## Endothelial IGF-1R overexpression alters the gut microbiota and augments the abundance of the beneficial genus *Akkermansia*

We then asked whether IGF-1R facilitated endothelial communication with the gut wall to influence the microbiota. Faith's phylogenetic diversity (PD), a measure of faecal microbial diversity, was significantly different in hIGFREO compared with WT after HFD (Fig 4A and B). Chao-1 analysis, a complementary measure of faecal microbial diversity and abundance, was also significantly different (Fig 4C and D). To further investigate these changes to the microbiota and assess the contribution of each *genus* to the difference between hIGFREO and WT, partial least squares discriminant analysis (PLS-DA) modelling and the variable importance in projection (VIP) score were performed. This demonstrated that hIGFREO mice on HFD have increased abundance of *Escherichia Shigella, Coriobacteriaceae UCG-002, Faecalibaculum, Peptococcus, Akkermansia* and *Dehalobacterium*. hIGFREO mice on a HFD are depleted in *Enterococcus, Barnesiella, Helicobacter, Streptococcs, Tyzzerella, Lachnospiraceae NK4A136* and *Bilophila,* as well as several genera from the Ruminococcaceae family (Fig 4E and F).

Of particular relevance to our findings was the increase in relative abundance of the genus *Akkermansia* (Derrien, 2004) seen in high-fat-fed hIGFREO. *Akkermansia* is thought to have anti-obesity and anti-diabetic effects in both humans and rodents (Everard *et al*, 2013; Cani & de Vos, 2017; Plovier *et al*, 2017; Depommier *et al*, 2019). Specifically, *Akkermansia muciniphila* reduces diet-induced weight gain, fat mass development, fasting hyperglycaemia and improves glucose tolerance without affecting food intake in mice (Everard *et al*, 2013), the same phenotype observed in hIGFREO. Increased levels of *Akkermansia muciniphila* are also associated with better clinical outcomes, such as insulin sensitivity, after a calorie restricted diet in overweight/obese adults (Dao *et al*, 2016). More recently, a proof-of-concept clinical trial in obese humans demonstrated that supplementation with *Akkermansia muciniphila* was a safe, well-tolerated intervention which improved several metabolic parameters (Depommier *et al*, 2019). However, it is also noteworthy that *Bilophila* was depleted in high-fat-fed hIGFREO; *Bilophila* has previously been shown to contribute to HFD-induced metabolic dysfunction (Natividad *et al*, 2018). *Dehalobacterium* was enhanced in high-fat-fed hIGFREO mice and has previously been shown to be protective against atherosclerosis and reduced cholesterol (Chan *et al*, 2016). It is difficult to speculate further about the contribution of these other genera as little more is known about their role in obesity and metabolic disease; further studies would be of interest. Interestingly, when hIGFREO mice were unchallenged on a chow diet, there was no difference in microbial diversity compared with WT (Fig EV5D–G).

To dissect potential mechanisms underpinning the altered microbial diversity, we examined the expression of genes known to modulate the microbiota (Chang & Kao, 2019). We saw several changes in gene expression in the gut wall (Fig EV5H–J), raising the possibility that crosstalk between endothelial cells and the gut wall can influence gene expression. It is well established that endothelial cells can act in a paracrine/autocrine fashion (Lee *et al*, 2007; Ding *et al*, 2010; Kivelä *et al*, 2019) and equally well established that enterocytes respond to microbial metabolites (Nuenen *et al*, 2005; Garrett, 2020). To examine a role for secreted factors from endothelial cells in the altered gene expression seen in hIGFREO small intestine, we used primary endothelial cells from hIGFREO to condition culture media to treat Caco-2 cells, as a model of the intestinal epithelial barrier. Caco-2 cells treated with conditioned media from hIGFREO showed a significant increase in regenerating islet-derived III-γ (REG3G) compared with WT gene expression (Fig EV5K). REG3G belongs to the family of C-type lectins and is one of several antimicrobial peptides produced by Paneth cells and enterocytes (Chang & Kao, 2019; Shin & Seeley, 2019). REG3G destroys gram-positive bacteria by binding to the peptidoglycan layer, exerting bactericidal activity by oligomerising to form hexameric transmembrane pores (Shin & Seeley, 2019), thus providing one explanation

Figure 4. Endothelial IGF-1R overexpression alters the gut microbiota and augments the abundance of the beneficial genus *Akkermansia*.

A, B   Faith's phylogenetic diversity (PD) was used to measure the faecal microbial diversity and demonstrates a significant difference between human IGF-1R endothelial overexpressing mice (hIGFREO) mice and wild-type littermates (WT) mice after high-fat diet feeding (*n* = 4–5 mice per group).

C, D   Chao-1 analysis was used to measure the faecal microbial diversity and abundance and demonstrates a significant difference between hIGFREO and WT (*n* = 4–5 mice per group).

E, F   Partial least squares discriminant analysis (PLS-DA) model and used the variable importance in projection (VIP) score was used to assess the contribution of each *genus*, shown as a scores plot in (E), and a loading plot of PLS-DA of genus abundances in (F). VIP score cut-off of 1 (*n* = 4–5 mice per group).

Data information: Data shown as mean ± SEM and individual mice are shown as data points. Diversity analyses were run on the resulting OTU/feature.biom tables to provide both phylogenetic and non-phylogenetic metrics of alpha and beta diversity. Additional data analysis (PLS-DA) and statistics were performed with R. *P* < 0.05 taken as being statistically significant using Student's *t*-test and denoted as *.

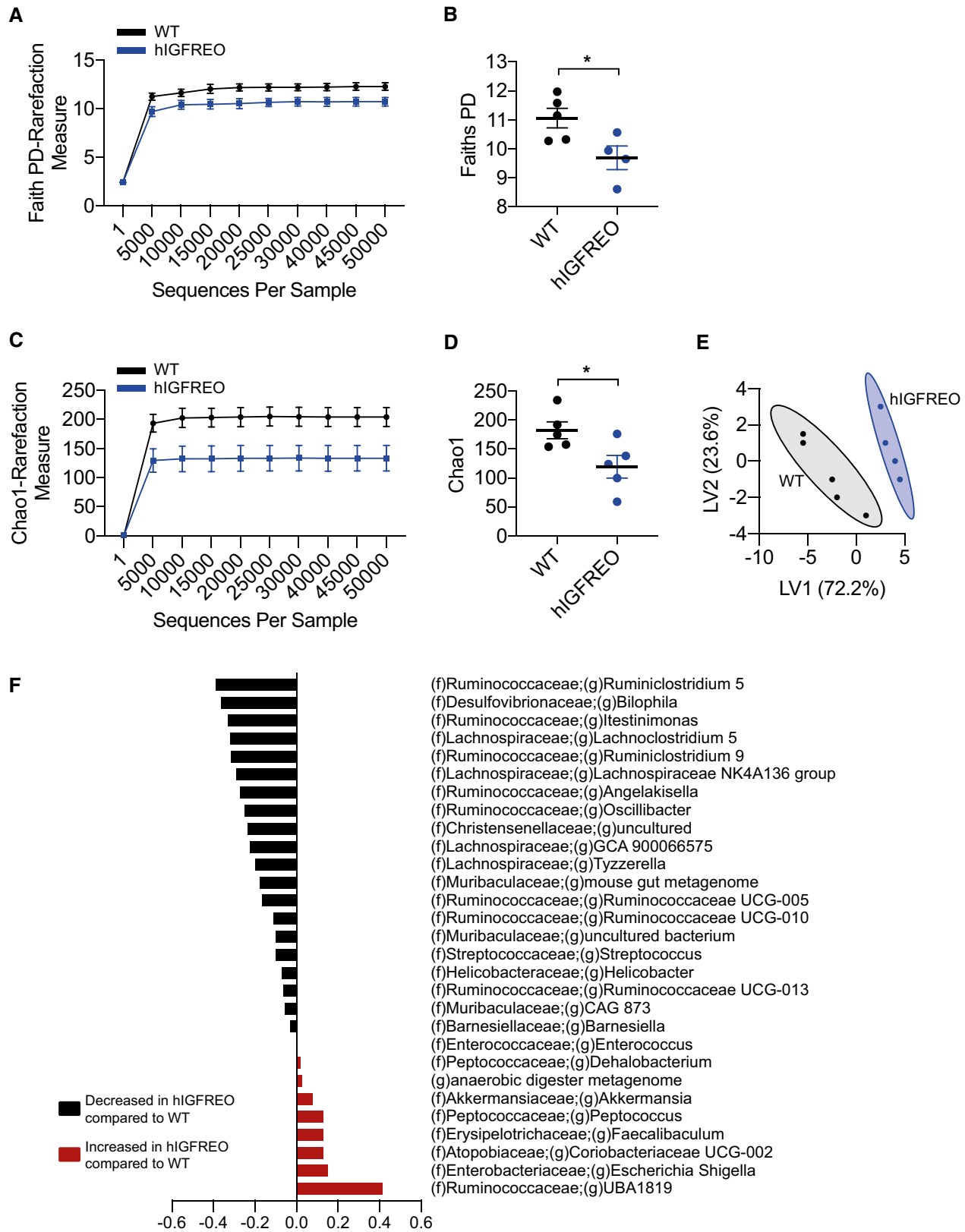

**Figure 4.**

as to why hIGFREO display reduced microbiota diversity and possibly providing an explanation as to why relative levels of *Akkermansia*, a gram-negative bacteria, are enhanced. This raises the intriguing possibility that endothelial cell IGF-1R could be a nutrient sensor responding to nutritional cues to influence the architecture of the intestinal microbiome (Bettedi & Foukas, 2017).

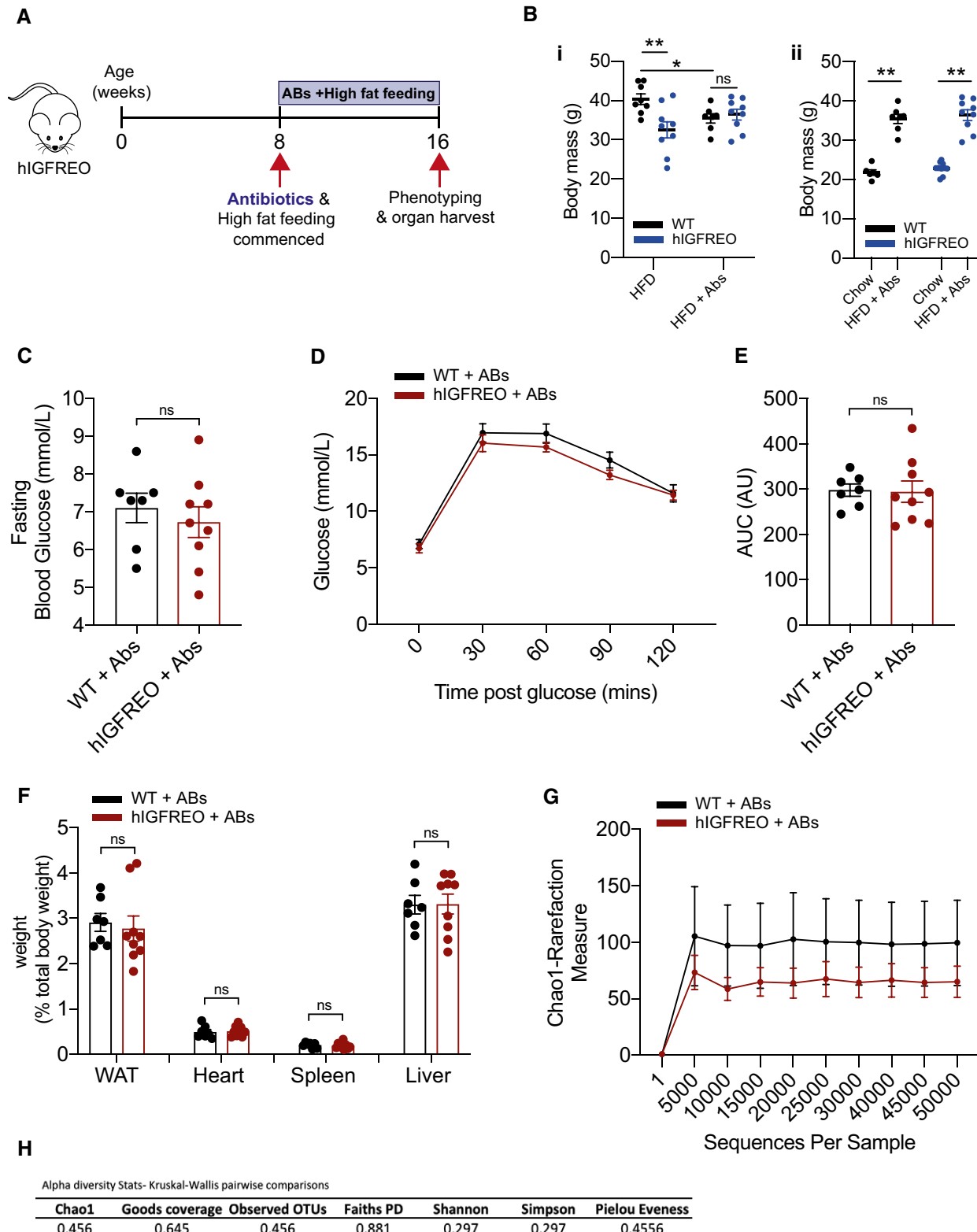

**Figure 5.**

◀

**Figure 5. Antibiotic administration in the setting of high-fat diet (HFD) eliminates the anti-obesity and anti-diabetic actions of endothelial IGF-1R overexpression.**

A     Schematic representation of antibiotic dosing and feeding time course.
B     (Bi), Human IGF-1R endothelial overexpressing mice (hIGFREO) had comparable weight gain as wild-type littermates (WT) after 8 weeks of HFD + antibiotics (ABs) when compared to WT. (Bii), Both hIGFREO and WT gained significant weight compared with chow-fed mice (*n* = 7–9 mice per group).
C     There was no difference in fasting blood glucose in hIGFREO compared with WT (*n* = 7–9 mice per group).
D, E   There was no difference in hIGFREO and WT glucose tolerance (as measured by glucose tolerance test and area under the curve (AUC)) (*n* = 7–9 mice per group).
F     Wet organ weights were similar in hIGFREO and WT (*n* = 7–9 mice per group).
G     Chao-1 analysis was used to measure the faecal microbial diversity and abundance and demonstrates no difference between hIGFREO and WT after HFD + antibiotic treatment (*n* = 3–5 mice per group).
H     Alpha diversity *P* values using Kruskal–Wallis pairwise comparisons show there is no difference in microbial diversity.

Data information: Data shown as mean ± SEM and individual mice are shown as data points, $P < 0.05$ taken as being statistically significant using Student's *t*-test and denoted as * or ** for *P* $P < 0.01$ and NS denotes not significant. Diversity analyses were run on the resulting OTU/feature.biom tables to provide both phylogenetic and non-phylogenetic metrics of alpha and beta diversity. Additional data analysis (PLS-DA) and statistics were performed with R.

## Antibiotic administration in the setting of obesity prevents the anti-obesity and anti-diabetic actions of endothelial IGF-1R overexpression

To investigate the contribution of the altered microbiota to the anti-obesity and anti-diabetic effects of endothelial IGF-1R overexpression, hIGFREO and WT were given broad-spectrum antibiotics in their drinking water (Rodrigues *et al*, 2017) for the duration of HFD (Fig 5A). The addition of antibiotic treatment alongside HFD abolished the difference in weight gain seen between hIGFREO and WT (Fig 5Bi). However, WT on HFD and antibiotic treatment did not gain as much weight as WT on HFD alone. On chow diet hIGFREO and WT did not tolerate prolonged antibiotic treatment and for welfare reasons had to be culled, thus suggesting that the mice did not completely tolerate antibiotic treatment. Nevertheless, both WT and hIGFREO gained significantly more weight than mice on chow diet (Fig 5Bii). Antibiotic treatment also prevented the difference in glucose intolerance, seen between the genotypes when on HFD alone (Fig 5C–E). Wet organ weights were comparable between hIGFREO and WT (Fig 5F). Chao-1 analysis was no different between hIGFREO and WT after HFD and antibiotic treatment (Fig 5G). Alpha diversity in hIGFREO and WT on HFD treated with antibiotics was also similar demonstrating no difference in microbial diversity using a range of approaches (Fig 5H and Table EV1). Taken together, these data confirm a causal role for the microbiota in the favourable changes seen in hIGFREO.

## Conclusion

To our knowledge, this is the first report to demonstrate communication between the endothelium and the gut wall, which in turn can modulate the gut microbiota. We report a novel role for endothelial cell IGF-1R in this crosstalk, which protects against diet-induced obesity and its associated adverse metabolic sequelae, by potentially remodelling the architecture of the microbiota.

# Materials and Methods

## Animal husbandry

hIGFREO mice with endothelial cell-specific overexpression of the IGF-1 receptor (previously described Imrie *et al,* 2012) and their wild-type control littermates (WT) were bred in house. Experiments were carried out under the authority of UK Home Office project licence P144DD0D6. Mice were group housed in cages of up to five, which contained a mix of genotypes. Researchers were blinded to genotype until the data analysis stage. Cages were maintained in humidity and temperature-controlled conditions (humidity 55% at 22°C) with a 12-h light–dark cycle. All interventions were performed within the light cycle. Only male mice were used for experimental procedures to prevent variability associated with the oestrous cycle on adiposity and metabolic readouts (Stubbins *et al,* 2012; Griffin *et al,* 2016). Genotyping was carried out by Transnetyx commercial genotyping using ear biopsies.

To induce obesity, mice received high-fat diet (HFD) *ad libitum* from 8 weeks of age for a further 8 weeks (60% of energy from fat) (F1850, Bioserve) with the following composition: protein 20.5, fat 36% and carbohydrate 36.2% (5.51 kcal/g).

Antibiotics were administered in a cocktail with the following concentrations: ampicillin (1/gl), metronidazole (1/gl), neomycin trisulfate (1/gl) and vancomycin (0.5/gl) in drinking water (Rodrigues *et al,* 2017), from the age of 8 weeks old for the duration of high-fat feeding (further 8 weeks).

## Metabolic phenotyping

Mice were fasted overnight prior to glucose tolerance or for 2hr prior to insulin tolerance tests. Blood glucose was measured using a handheld Glucose Meter (Accu-Chek Aviva). An intra-peritoneal injection of glucose (1 mg/g) or recombinant human insulin (Actrapid; Novo Nordisk) (0.75 IU/kg) was given and glucose concentration measured at 30-min intervals for 2 h from the point of glucose/insulin administration. Mice were not restrained between measurements (Haywood *et al,* 2017).

Fasting plasma samples were collected from the lateral saphenous vein (EDTA collection tubes Sarstedt 16.444). Samples were then spun at 12,300 *g* for 10 min in a bench top centrifuge. Fasting plasma insulin (90080, Crystal Chem), IGF-I (MG100, R and D systems), leptin (EZML-82K, Merck Millipore) and adiponectin (EZMADP-60K, Merck Millipore) were measured as per manufacturer's instructions.

Core body temperature was measured using an Indus rectal temperature probe (Vevo2100 (VisualSonics, FujiFilm).

After 8 weeks of HFD, metabolic parameters were measured by indirect calorimetry using Comprehensive Lab Animal Monitoring

Systems (CLAMS) (Columbus Instruments). In brief, mice were individually housed for 5 days and measurement of oxygen consumption, carbon dioxide production, food intake and locomotor activity were continuously recorded. For each mouse, a full 24-h period, taking into account sleep and wake cycles, was analysed after an acclimatisation period (Roberts *et al,* 2014).

After 8 weeks of HFD (or at 8 weeks old for chow control mice), all mice were sacrificed using terminal anaesthesia and organ weights measured using a standard laboratory balance.

### Lipid absorption

Mice were fasted overnight and blood samples collected from the lateral saphenous vein (EDTA collection tubes Sarstedt 16.444). Mice underwent oral gavage with 200 μl olive oil, and blood was taken from the saphenous vein every hour for a further 3 h (Zhang *et al,* 2018). Plasma triglycerides were measured using a commercially available kit (ab65336, Abcam).

### Intestinal transit time

Mice were fasted overnight before oral gavage with 300 μl of Carmine solution (6% Carmine red (C1022, Sigma) in 0.5% methyl cellulose (M7140, Sigma-Aldrich)). Mice were then individually caged and monitored until the appearance of the first red faecal pellet (Li *et al,* 2011).

### Magnetic resonance imaging (MRI)

Anaesthesia was induced using 5% isoflurane in 100% oxygen and then maintained using 1.5–3% isoflurane at 2 l/min oxygen flow. Animals were positioned prone on a dedicated mouse cradle. Body temperature was maintained with a custom resistive blanket placed on the back of the animal. Cardiac and respiratory signals were continuously monitored (BIOPAC Systems, Inc., Goleta, USA). Mice were imaged on a 7T preclinical MRI scanner with a 660 mT/m shielded gradient system and a quadrature-driven transmit/receive volume coil with inner diameter of 72 mm (Bruker BioSpin MRI GmbH, Ettlingen, Germany). A 2D cardiac-triggered and respiratory-gated 3-point Dixon spoiled gradient-echo sequence was used: TR = 5.65 ms, TE = 2.42/2.75/3.09 ms, Matrix = 256 × 128, field-of-view = 80 × 30 mm, number of slices = 28 in sagittal orientation, slice thickness = 1 mm, number of signal averages = 8, total scan time ~30 min. The data were analysed in MATLAB (Math-Works, Natick, USA) using the hierarchical iterative decomposition of water and fat with echo asymmetry and least squares estimation (IDEAL) method(Tsao & Jiang, 2013). The proton density fat fraction (PDFF, the amount of lipid signal over total signal) was used to segment adipose tissue depots. Subcutaneous and visceral adipose depots were segmented separately using Osirix Lite v11.0.2 (Bernex, Switzerland) 2D threshold region growing algorithm tool with segmentation parameters set to a lower threshold of 80% PDFF.

### Gene expression

RNA was isolated from cells and tissue samples using the monarch total RNA mini kit (NEB, T2010S). The concentration of RNA in each sample (ng/μl) was measured using a NanoDrop. cDNA was

reverse transcribed (NEB, E3010S). Quantitative PCR (qPCR) was performed using a Roche LightCycler 480 Instrument II, using SYBR Green PCR Master Mix (Bio-Rad, 1725270) and relevant primers (See Table 1). The "cycles to threshold" (cT) was measured for each well, the average of triplicate readings for each sample taken, normalised to GAPDH, and finally, the differential expression of each gene was calculated for each sample.

### Quantification of protein expression

Cells were lysed or tissue mechanically homogenised in lysis buffer (Extraction buffer, FNN0011) and protein content quantified using a BCA assay (Sigma-Aldrich, St. Louis, MO). Twenty micrograms of protein was resolved on a 4–12% Bis-Tris gel (Bio-Rad, Hertfordshire, UK) and transferred to nitrocellulose membranes. Membranes were probed with antibodies diluted in 5% BSA as per Table 2, before incubation with appropriate secondary horseradish peroxidase-conjugated antibody. Blots were visualised with Immobilon Western Chemiluminescence HRP Substrate (Merck Millipore, Hertfordshire, UK) and imaged with Syngene chemiluminescence imaging system (SynGene, Cambridge, UK). Densitometry was performed in ImageJ (Haywood *et al,* 2017).

### Primary endothelial cell isolation

Primary endothelial cells (PECs) were isolated from lungs, as previously reported (Abbas *et al,* 2011; Watt *et al,* 2017). Briefly, lungs were harvested, washed, finely minced and digested in Hanks' balanced salt solution containing 0.18 units/ml collagenase (10 mg/ml; Roche) for 45 min at 37°C. The digested tissue was filtered through a 70-μm cell strainer and centrifuged at 400 *g* for 10 min. The cell pellet was washed with PBS/0.5%BSA, centrifuged, resuspended in 1 ml PBS/0.5% and incubated with $1 \times 10^6$ CD146 antibody-coated beads (Miltenyi Biotec, 130-092-007) at 4°C for 30 min. Bead-bound endothelial cells were separated from non-bead-bound cells using a magnet.

### Microbiome analysis

Microbiome analysis was performed by UC Davis MMPC. Briefly, frozen faecal samples were shipped on dry ice to UC Davis MMPC and Host Microbe Systems Biology Core. Total DNA was extracted using Mo-Bio (now Qiagen) Power Fecal Kit. Sample libraries were prepared and analysed by barcoded amplicon sequencing(Anderson, 2001; Price *et al,* 2010; Lozupone *et al,* 2011; Quast *et al,* 2012; Katoh & Standley, 2013; Mandal *et al,* 2015; Callahan *et al,* 2016; Bolyen *et al,* 2019).

### Quantification of white and brown adipose tissue vascularity

White adipose tissue (WAT) and brown adipose tissue (BAT) (< 0.5 g) were harvested into cold 1% paraformaldehyde (PFA) and allowed to fix for 2hrs at room temperature. Samples were incubated overnight with Isolectin B4 Alexa Fluor 647 (I32450, Thermo Fisher Scientific) and diluted 1:100 in 5% BSA in phosphate-buffered saline (PBS) at 4°C. After washing with PBS, they were incubated with HCS LipidTOX (H34475, Thermo Fisher Scientific) diluted 1:200 in PBS for 20mins at room temperature. Whole tissue

**Table 1. Primer details for qPCR.**

| Gene | | Assay ID |
|------|------|----------|
| IGF1R | Insulin-like growth factor-1 receptor | qHsaCED0044963 |
| PIGR | Polymeric immunoglobulin receptor | qHsaCID0021506 |
| MMP7 | Matrix metallopeptidase 7 | qHsaCED0044775 |
| REG3G | Regenerating family member 3 gamma | qHsaCED0004912 |
| NLRP6 | NLR family pyrin domain containing 6 | qHsaCED0004389 |
| GAPDH | Glyceraldehyde-3-phosphate dehydrogenase | qHsaCED0038674 |
| Insr | Insulin receptor | qMmuCID0018034 |
| Ucp1 | Uncoupling protein 1 | qMmuCID0005832 |
| Pparg | Peroxisome proliferator-activated receptor gamma | qMmuCID0018821 |
| Ppargc1a | Peroxisome proliferative-activated receptor, gamma, coactivator 1 alpha | qMmuCID0006032 |
| Vegfa | Vascular endothelial growth factor A | qMmuCED0040260 |
| Adipoq | Adiponectin | qMmuCID0023242 |
| Lep | Leptin | qMmuCID0040177 |
| Cyp7a1 | Cytochrome P450, family 7, subfamily a, polypeptide 1 | qMmuCED0046994 |
| Abcb11 | ATP-binding cassette, subfamily B (MDR/TAP), member 11 | qMmuCID0015514 |
| Mttp | Microsomal triglyceride transfer protein | qMmuCED0047210 |
| Apob | Apolipoprotein B | qMmuCED0044141 |
| Plagl2 | Pleiomorphic adenoma gene-like 2 | qMmuCED0037791 |
| Sar1b | Secretion-associated Ras-related GTPase 1B | qMmuCED0044630 |
| Acsl3 | Acyl-CoA synthetase long-chain family member 3 | qMmuCED0046845 |
| Acsl5 | Acyl-CoA synthetase long-chain family member 5 | qMmuCED0045475 |
| Atl1 | Atlastin GTPase 1 | qMmuCED0044156 |
| Cideb | Cell death-inducing DFFA-like effector b | qMmuCED0046272 |
| Dgat1 | Diacylglycerol O-acyltransferase 1 | qMmuCID0021210 |
| Dgat2 | Diacylglycerol O-acyltransferase 2 | qMmuCID0012338 |
| Mgat2 | Mannoside acetylglucosaminyltransferase 2 | qMmuCED0049876 |
| Plin2 | Perilipin 2 | qMmuCID0016776 |
| Plin3 | Perilipin 3 | qMmuCID0005622 |
| Aqp7 | Aquaporin 7 | qMmuCID0025269 |
| Tlr3 | Toll-like receptor 3 | qMmuCID0005723 |
| Tlr5 | Toll-like receptor 5 | qMmuCID0005789 |
| Nod2 | Nucleotide-binding oligomerisation domain containing 2 | qMmuCED0049905 |
| Nlrp6 | NLR family pyrin domain containing 6 | qMmuCED0048619 |
| Clec1b | C-type lectin domain family 1, member b | qMmuCED0047632 |
| Lct | Lactase | qMmuCED0045800 |
| Ahr | Aryl-hydrocarbon receptor | qMmuCED0044800 |

**Table 1 (continued)**

| Gene | | Assay ID |
|------|------|----------|
| Nr1h4 | Nuclear receptor subfamily 1, group H, member 4 | qMmuCID0014006 |
| Vdr | Vitamin D receptor | qMmuCID0006555 |
| Aldh1l1 | qMmuCID0021991 | qMmuCID0021991 |
| Reg3g | Regenerating family member 3 gamma | qMmuCED0040314 |
| Retnlb | Resistin-like beta | qMmuCED0001569 |
| Defb1 | Defensin beta 1 | qMmuCID0008786 |
| Mmp7 | Matrix metallopeptidase 7 | qMmuCID0022398 |
| F11r | F11 receptor | qMmuCID0006275 |
| Myo1a | Myosin IA | qMmuCID0022137 |
| Mgam | Maltase-glucoamylase | qMmuCID0022182 |
| Pigr | Polymeric immunoglobulin receptor | qMmuCID0009049 |
| Ifng | Interferon gamma | qMmuCID0006268 |
| Il4 | Interleukin 4 | qMmuCID0006552 |
| Muc2 | Mucin 2 | qMmuCID0019583 |
| Muc3 | Mucin 3 | qMmuCID0023019 |
| Flt1 | FMS-like tyrosine kinase 1 | qMmuCID0016762 |
| Kdr | Kinase insert domain protein receptor | qMmuCID0005890 |
| Flt4 | FMS-like tyrosine kinase 4 | qMmuCID0021117 |
| Vegfc | Vascular endothelial growth factor C | qMmuCID0017182 |
| Cd36 | CD36 antigen | qMmuCID0014852 |
| Gapdh | Glyceraldehyde-3-phosphate dehydrogenase | qMmuCED0027497 |

**Table 2. Antibody details for protein expression.**

| Protein | Supplier | Code | Secondary antibody |
|---------|----------|------|--------------------|
| IGF1R (D23H3) | Cell Signaling | #9750 | Anti Rabbit |
| IR (4B8) | Cell Signaling | #3025 | Anti Rabbit |
| AKT | Cell Signaling | #9272 | Anti Rabbit |
| p-AKT Ser473 | Cell Signaling | #9271 | Anti Rabbit |
| eNOS | Cell Signaling | #9572 | Anti Rabbit |
| p-eNOS Ser 1177 | Cell Signaling | #9570 | Anti Rabbit |

was then mounted onto slides beneath coverslips using a silicone spacer (Grace bio-labs, 664113), with Prolong Gold (P36930, Thermo Fisher Scientific). Slides were then imaged using laser scanning confocal microscopy (LSM880, Zeiss), with 8 areas of each sample imaged. Vascular density (the proportion of each image stained with IB4) was measured using thresholding in ImageJ.

**Histological assessment of adipocyte size, non-alcoholic fatty liver disease and villi structure**

Samples for histology were fixed in 4% PFA for at least 24 h and then processed into paraffin blocks. 5-μm sections were taken and collected onto 3-triethoxysilylpropylamine (TESPA) coated slides.

After drying, slides were stained with haematoxylin and eosin to assess gross morphology ± oil red o (ORO) for lipid staining. Slides were imaged using an Olympus BX41 microscope at 10× and 20× magnification.

For assessment of adipocyte size, three separate fields of view for each sample were assessed. For each one, the average of 20 randomly selected independent cells measured using ImageJ.

For assessment of non-alcoholic fatty liver disease (NAFLD) in sections of murine liver, a validated rodent NAFLD scoring system was used (Liang *et al*, 2014), which takes into account micro- and macro-steatosis, inflammation and hypertrophy. Each sample was assessed by at least two blinded independent verifiers (NH, KB or NW) and the average score per sample taken.

### Flow cytometry

To isolate the stromal vascular fraction, epididymal fat pads were harvested, washed, finely minced and digested in Hanks' balanced salt solution containing collagenase (1 mg/ml; Roche) for 45 min at 37°C. The digested tissue was agitated using a cannula and centrifuged at 1,000 rpm for 10 min. The upper lipid phase was removed and the aqueous phase with pellet was filtered through a 70-μM cell strainer and centrifuged at 1,000 rpm for 7 min. The pellet was resuspended in PBS containing 0.5% BSA (Sigma-Aldrich) and 2 mM EDTA (Sigma-Aldrich) and was filter through a 30-μM cell strainer and further centrifuged at 1,000 rpm for 7 min.

Cells from the stromal vascular fraction were washed and resuspended in PBS containing 0.5% BSA (Sigma-Aldrich) and 2 mM EDTA (Sigma-Aldrich). Fc receptors were blocked with a CD16/32 antibody (Miltenyi Biotec, 130-092-575) for 10 min at 4°C. Samples were then incubated with anti-CD45-VioBlue (Miltenyi Biotec, 130-110-802), anti-CD11b-FITC (Miltenyi Biotec, 130-081-201), anti-Ly6G-PE (Miltenyi Biotec, 130-107-913), anti-Ly6C-APC (eBioscience, 17-5932-82) or anti-F4/80-APC (Miltenyi Biotec, 130-102-379) for 10mins at 4°C, according to the manufacturer's protocol. Stained cells were washed in PBS containing 0.5% BSA and 2 mM EDTA. Samples were analysed by flow cytometry (CytoFLEX S, Beckman Coulter). Leukocytes were identified based on typical light scatter properties, with further gating to define: $CD45^+$ leukocytes, $CD45^+CD11b^+$ myeloid cells, $CD45^+CD11b^+Ly6G^-Ly6C^{hi}$ inflammatory monocytes, $CD45^+CD11b^+Ly6G^-Ly6C^{low}$ reparative monocytes, $CD45^+CD11b^+Ly6G^{hi}Ly6C^{hi}$ neutrophils and $CD45^+CD11b^+F4/80^+$ macrophages. Data were scaled to cells/ml of blood or weight of fat pad.

### Pancreatic lipase activity

Tissue was harvested under terminal anaesthesia. 40 mg of pancreas was homogenised and used in a lipase activity assay (Abcam, ab102524).

### Liver and plasma lipid measurements

100mg of tissue was weighed and homogenised in 1 ml of 5% Igepal (I8896, Sigma) and heated to 80°C for 5 min, cooled and reheated again before centrifuging for 2 min. The supernatant was used to measure, triglycerides, free fatty acids and cholesterol (Abcam, ab65336, ab65341 and ab65359, respectively).

### Conditioning media

Conditioned media experiments require a large number of EC, and pulmonary EC provides an appropriate yield of cells to perform these experiments. Therefore, when PECs reached confluency, supplemented growth media was removed and replaced with basal endothelial growth medium-MV2 for 24 h. Conditioned media was then removed and used in further experiments as described.

### Caco-2 cells

Caco-2 cells were purchased from Public Health England Culture Collections (86010202). Pre-differentiated Caco-2 cells were maintained in 20% (v/v) FBS/MEME containing non-essential amino acids, 0.292 g/l L-glutamine, 2.2 g/l sodium bicarbonate (Sigma-Aldrich #M4655), supplemented with 1XAntibiotic Antimycotic Solution (Sigma-Aldrich #A5955) and incubated at 37°C in 5% $CO_2$. Upon confluency, Caco-2 cell differentiation was initiated by seeding on Transwell inserts. Differentiated Caco-2 cells were subjected to boiled (5 min at 95°C) conditioned media stimulation at the basal side for 24 h at 37°C in 5% CO2.

### Data analysis

All data are shown as mean ± standard error of mean (SEM) unless stated, with individual mice presented as data points. All image analysis was performed in ImageJ unless stated. Student unpaired *t*-test was used for statistical analyses and performed with GraphPad Prism software version 8 unless stated. For plasma concentration–time profile experiments, area under the curve analyse was used and performed with GraphPad Prims. For metabolic parameters measured by indirect calorimetry, ANOVA testing was performed using mass as a co-variant (ANCOVA testing) using calrapp.org. $P < 0.05$ taken as statistically significant.

## Data availability

No primary data sets have been generated and deposited.

**Expanded View** for this article is available online.

### Acknowledgements

NJH was funded by British Heart Foundation Project grant (PG/18/82/34120). CL was funded by a British Heart Foundation studentship (FS/19/59/34896). MD was funded by British Heart Foundation Clinical Research Training Fellowship (FS/18/44/33792). LDR was funded by a Diabetes UK RD Lawrence Fellowship (16/0005382). RMC was funded by a British Heart Foundation Clinical Intermediate Fellowship (FS/12/80/29821). MTK holds a British Heart Foundation Chair in Cardiovascular and Diabetes Research (RG/15/7/31521). NTW was funded by a British Heart Foundation project grant (PG/14/54/30939). The Experimental and Preclinical Imaging Centre was co-funded by the BHF (SI/14/1/30718). We would like to acknowledge the histology service from the Division of Pathology and Data Analytics, Colorectal Pathology Trials, University of Leeds, for sectioning and staining adipose and liver samples. We would also like to acknowledge the Bio-imaging and Flow Cytometry Facility, Faculty of Biological Sciences, University of Leeds, for acquiring images from histology slides. We would also like to acknowledge the UC Davis MMPC

services, whose research was supported by NIH grant U24-DK092993 (MMPC-University of California Davis Microbiome and Host Response Core, RRID:SCR_015361). MTK is the guarantor of this work and, as such, had full access to all the data in the study and takes responsibility for the integrity of the data and the accuracy of the data analysis.

## Author contributions

NJH, KIB, NM, AS, NYY performed *in vivo* experiments. NJH, CL, KIB, CHO, NW, MD, CGW, NTW performed *ex vivo* experiments. JK-P, IT, JHB, SS, JES, AM performed *in vivo* imaging experiments. NJH, CL, performed cell culture experiments. NJH and MTK wrote the manuscript. NJH, CL, KIB, NM, AS, CHO, NW, MD, CGW, NTW, JKP, IT, JHB, SS, JES, NYY, LDR, DJB, PS, SBW, RMC, MTK reviewed the manuscript. MTK, RMC, SBW, NYY, LDR, DJB, PS obtained funding.

## Conflict of interest

The authors declare that they have no conflict of interest.

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
