## [Review Process File · EMBO Reports]

Endothelial IGF-1 receptor mediates crosstalk with the gut wall to regulate microbiota in obesity

Natalie Haywood, Cheukyau Luk, Katherine Bridge, Michael Drozd, Natallia Makava, Anna Skromna, Amanda Maccannell, Claire Ozber, Nele Warmke, Chloe Wilkinson, Nicole Watt, Joanna Koch-Paszkowski, Irvin Teh, Jordan Boyle, Sean Smart, Jurgen Schneider, Nadira Yuldasheva, Lee Roberts, David Beech, Piruthivi Sukumar, Stephen Wheatcroft, Richard Cubbon, and Mark Kearney
DOI: [10.15252/embr.202050767](https://doi.org/10.15252/embr.202050767)

Corresponding author(s): Mark Kearney (m.t.kearney@leeds.ac.uk)

Review Timeline:

Submission Date:	28th Apr 20
Editorial Decision:	26th May 20
Revision Received:	1st Feb 21
Editorial Decision:	23rd Feb 21
Revision Received:	12th Mar 21
Accepted:	22nd Mar 21

Editor: Achim Breiling

Transaction Report:

Dear Dr. Haywood,

Thank you for the submission of your research manuscript to EMBO reports. We have now received reports from the three referees that were asked to evaluate your study, which can be found at the end of this email.

As you will see, all referees think that the findings are of interest, but they also have several comments, concerns and suggestions, indicating that a major revision of the manuscript is necessary to allow publication in EMBO reports. As the reports are below, and I think all points need to be addressed, I will not detail them here. It will be of particular importance, though, to add the control groups as indicated by referee #3.

Given the constructive referee comments, we would like to invite you to revise your manuscript with the understanding that all referee concerns must be addressed in the revised manuscript and in a detailed point-by-point response. Acceptance of your manuscript will depend on a positive outcome of a second round of review. It is EMBO reports policy to allow a single round of revision only and acceptance of the manuscript will therefore depend on the completeness of your responses included in the next, final version of the manuscript.

Revised manuscripts should be submitted within three months of a request for revision. We are aware that many laboratories cannot function at full efficiency during the current COVID-19/SARS-CoV-2 pandemic and we have therefore extended our 'scooping protection policy' to cover the period required for full revision. Please contact me to discuss the revision should you need additional time, and also if you see a paper with related content published elsewhere.

1) a .docx formatted version of the final manuscript text (including legends for main figures, EV figures and tables), but without the figures included. Please make sure that changes are highlighted to be clearly visible. Figure legends should be compiled at the end of the manuscript text.

Please order the manuscript sections like this:

Title page - Abstract - Introduction - Results - Discussion - Materials and Methods - DAS - Acknowledgements - Author contributions - Conflict of interest - References - Figure legends - Expanded View Figure legends

2) individual production quality figure files as .eps, .tif, .jpg (one file per figure), of main figures and EV figures. Please upload these as separate, individual files upon re-submission.

For more details please refer to our guide to authors:

See also our guide for figure preparation:

http://wol-prod-cdn.literatunonline.com/pb-assets/embosite/EMBOPress_Figure_Guidelines_061115-1561436025777.pdf

4) a complete author checklist, which you can download from our author guidelines (<https://www.embopress.org/page/journal/14693178/authorguide>). Please insert page numbers in the checklist to indicate where the requested information can be found in the manuscript. The completed author checklist will also be part of the RPF.

Please also follow our guidelines for the use of living organisms, and the respective reporting guidelines: <http://www.embopress.org/page/journal/14693178/authorguide#livingorganisms>

5) that primary datasets produced in this study (e.g. RNA-seq, ChIP-seq and array data) are deposited in an appropriate public database. This is now mandatory (like the COI statement). If no primary datasets have been deposited in any database, please state this in this section (e.g. 'No primary datasets have been generated and deposited').

The accession numbers and database should be listed in a formal "Data Availability " section (placed after Materials & Methods) that follows the model below. Please note that the Data Availability Section is restricted to new primary data that are part of this study.

Data availability

- RNA-Seq data: Gene Expression Omnibus GSE46843

(<https://www.ncbi.nlm.nih.gov/geo/query/acc.cgi?acc=GSE46843>)

- [data type]: [name of the resource] [accession number/identifier/doi] ([URL or

identifiers.org/DATABASE:ACCESSION])

6) We strongly encourage the publication of original source data with the aim of making primary data more accessible and transparent to the reader. The source data will be published in a separate source data file online along with the accepted manuscript and will be linked to the relevant figure. If you would like to use this opportunity, please submit the source data (for example scans of entire gels or blots, data points of graphs in an excel sheet, additional images, etc.) of your key experiments together with the revised manuscript. If you want to provide source data, please include size markers for scans of entire gels, label the scans with figure and panel number, and send one PDF file per figure.

8) Regarding data quantification and statistics, can you please specify, where applicable, the number "n" for how many independent experiments (biological replicates) were performed, the bars and error bars (e.g. SEM, SD) and the test used to calculate p-values in the respective figure legends. Please provide statistical testing where applicable, and also add a paragraph detailing this to the methods section. See: <http://www.embopress.org/page/journal/14693178/authorguide#statisticalanalysis>

9) Please provide a shorter title, with not more than 100 characters (including spaces).

10) Please remove the abbreviations section from the manuscript text. Please define each abbreviation when it is mentioned in the text for the first time.

I look forward to seeing a revised version of your manuscript when it is ready. Please let me know if you have questions or comments regarding the revision.

Yours sincerely

Achim Breiling
Editor
EMBO Reports

Referee #1:

In this study, Haywood et al investigated the influence of ECs on the gut microbiota using an

hIGFREO mice. The major findings include the improved glucose tolerance, reduced adiposity, and reduced body weight in hIGFREO mice under HFD as compared to the wildtype littermates. An altered microbial diversity was observed in hIGFREO mice under HFD. Of particular interest, Akkermansia was found to be increased. To prove that the difference in microbiota is an underlying mechanism of the observed phenotype, it was shown that the administration of broad-spectrum antibiotics minimized the difference between wildtype and hIGFREO. Additionally, hIGFREO ECs conditioned media increased REGIII α in Caco-2 cells which provides clues for EC-gut crosstalk. Overall, this study provides interesting data connecting EC and enterocytes through possible modulation of gut microbiota. These data could provide novel insights into the important regulation of intestinal functions and metabolic states by the endothelium. A number of issues need to be addressed to strengthen the arguments and consolidate the conclusion.

Major concerns:

1. The data from adipose tissues, which show the major difference at the organ level, need to be more carefully examined and interpreted:
 - In Fig. S2B, authors should quantify the numbers and sizes of adipocytes and present data using histograms to show the distribution, rather than using adipocyte area. According to Fig. S2A, there appear to be more adipocytes of smaller sizes in hIGFREO as compared to the wildtype.
 - In Fig. S2C, IB4 staining in BAT is apparently different between wildtype and hIGFREO. Furthermore, Fig. S2F-H show trendy decrease of leucocyte numbers in the WAT. Scatter plot should be shown to demonstrate the distribution of the data. It is not clear the representative data are based on how many mice examined.
 - It was claimed that "adipose tissue expression of browning markers were not different in hIGFREO compared to WT (Figure 3D)". However, in Fig. 3D, Pgc1 α (possibly misspelled for Pgc1a?) is apparently increased in the WAT. Same markers (Ucp-1, pgc1a, vegfa, adipoq, and lep) should be examined in both WAT and BATs.
2. The primary mechanism underlying the anti-obese phenotype in hIGFREO has been attributed to changes in the gut, which should be corroborated. At least, histology should be shown for the intestines from the wildtype and the hIGFREO mice.
3. It is interesting that the antibiotics abolished the observed difference between the wildtype and hIGFREO. Data supporting that the EC overexpression of IGFR works through microbiota need to be consolidated:
 - In Fig. 5B, WT with or without ABs, and hIGFREO with or without ABs should be plotted on the same graph to enable the demonstration of the effect of ABs per se on body mass. It seems that WT + ABs already have lower body mass than WT without ABs (in Fig. 1D). This needs to be addressed as premise before further interpretation between WT and hIGFREO, both administered ABs.
 - In Fig. 6A and B, using heatmap to present the qPCR data is uncommon and fails to reflect the quantitative nature of qPCR and the variations among individual samples. Scatter bar plots should be used instead of heatmaps. Also, correction for multiple hypothesis testing needs to be performed for these data, rather than a simple t test.
 - In Fig. 6C, the comparison should be done between hIGFREO vs WT, rather than basal media.
4. To explain how hIGFREO change the intestinal gene expression and the consequent microbiota, data from intestinal ECs needed be provided. Markers for barrier function and immune response should be measured. IGFR should also be determined at the protein level since Tie2 is not entirely specific for ECs.
5. The last result section needs clarification. There are 6 experimental groups and the rationale for comparisons is not clear. For example, Fig. 6B "demonstrated an upregulation of Reg3g, Mmp7, Nlrp6 and Pigr". This is an incomplete statement- upregulation in which group? Similarly, "Gene expression of Caco-2 cells treated with conditioned media from hIGFREO endothelial cells showed

a significant increase in Reg3g gene expression (Figure 6C), as seen in high fat fed hiGFREO mice (compared to what?).

6. The numbers of mice used are not specified for any of the experiments. This should be clearly indicated for each of the experiment. For ANOVA, posthoc test should be specified.

7. A justification for only using male mice should be provided.

Minor points:

1. In Fig. 1D, data should be presented in the same manner as in Fig. 5D, i.e. should measurements in the time course.

2. In figure 4D, is this difference significant? Description of data showing trends toward difference without statistical significance should be made more accurate, rather than simply stating "no change/no difference".

3. Supplemental Table 1 does not seem to be included in the submission. Please provide.

4. For Fig. S2C, the color assignment is inconsistent between figure legend and the images. In Fig. S2A-C, same tissues should be assayed, including iWAT, eWAT, and BAT.

5. Discussion from Line 249-260 is puzzling. Authors should provide a reasonable conclusion for these findings in the context of the current work.

6. "It has been shown that Akkermansia muciniphila can increase Reg3g expression in mice fed a standard chow diet, but not high fat, further suggesting that hiGFREO mice have a beneficial phenotype when challenged, which is usually diminished by high fat feeding." This seems contradictory to Fig. 6B showing Reg3g still increased in hiGFREO under HFD with ABs. Please clarify.

7. Whenever referring to specific data, figures should be cited in discussion.

8. Authors should indicate the rationale of using Caco-2 cells.

9. Please proofread carefully to avoid any typos and mis-labeling, e.g. Line 171 "paracrine manor (manner?)."

Referee #2:

Haywood et al. utilized transgenic mice with overexpressing human IGF-1R in endothelial cells showing the IGF-1R effects on adiposity reduction, glucose tolerance, and also fecal microbiota composition. They proposed the IGF-1R facilitates the crosstalk between endothelial cells and gut microbiota, which protects against diet-induced obesity. The data are solid and findings are importance.

IGF-1R overexpressed mice have a lower gut microbiota diversity (lower PD) than wild type mice (Figure 4). As a complementary confirmation of their observation that IGF-1R on obesity through gut microbiota, they should consider using some way like fecal transfer of wild type mice to the IGF-1R mice to reverse gut microbiota compositions and then evaluate its effect.

Applying the antibiotics cocktails depleted all gut microbiota, further decreased microbiota PD diversity, and not only depleted "beneficial bacterial taxa", but also "bad bacterial taxa", which may generate multiple effects on host physiology and may not support their conclusions on the crosstalk.

The microbiota sequencing analysis based on the text only shows relative abundances, not the real abundance. So if they claim hiGFREO increases Akkemansia (as potential key player), they should use qPCR to compare total 16S and Akkemansia 16S abundance between wild type and

transgenic mice.

Referee #3:

Haywood et al., examined the effects of endothelial IGF-1R overexpression on metabolic function in experimental mice. They report that high fat diet-induced glucose intolerance is mitigated in IGF-1R mice, and this effect is due to alterations in the gut microbiota.

The study is well conceived and conducted. However, there are some concerns and questions regarding the data, and some limitations that reduce excitement for the manuscript.

- first and foremost, the experiments are lacking proper control groups. Specifically, WT and IGFR mice on a control diet (and use of Abx on a control diet in the second experiment). These groups are necessary to adequately analyze the effects of the diet and the overexpression.

-The conclusions derived by the authors are not fully supported by the data. For example, the change in Akkermansia is rather small (a control group would help here), other bacterial species were altered to a much greater extent, so the relative importance of Akkermansia is questioned. Further, the link to REG3 is tenuous, and the importance/necessity of REG3 is based only on gene expression

-Introduction: The introduction does not sufficiently explain why the authors hypothesized that IGF1R overexpression would improve metabolic function, or why the microbiota would be involved. Are there existing data that provided a scientific rationale for the study?

-Most metabolic outcomes were not different in the IGFR mice, glucose tolerance appeared to be an exception. For that reason, tissue data providing more mechanistic insight into the difference would strengthen the data.

-Given the site of overexpression, it would be appropriate to examine markers of vascular/endothelial function.

-Can the authors explain why pulmonary ECs were isolated? Are these cells phenotypically similar to more commonly examined ECs (e.g. aortic; peripheral microvascular)

-Regarding antibiotic treatment, how were they administered? Also, a universal primer would confirm the success of the antibiotic treatment. Did the mice tolerate the Abx treatment well? Several studies have reported that mice do not tolerate Abx treatment, and sugar must be added to encourage feeding. The magnitude of weight gain in the Abx experiment looks smaller than the first experiment, which could support this notion (again a control group would help here).

- Can the authors explain the significance of the lipid absorption test? And why 3 hours? I would have thought TG are still being produced at 3h, and many postprandial tests measure out 4 to 6 hours.

Dr Achim Breiling
Editor
EMBO Reports

Dear Dr Breiling,

We would like to thank you and the referees for the fair and constructive reviews of our manuscript, which we feel we have been able to respond to in full.

After completing a substantial amount of extra work, in particular in mice fed a chow diet and in female mice we think the revised manuscript now reaches the novelty and exacting standards expected by the editorial board and readership of EMBO Reports. Our point-by-point responses to the editor and reviewers are detailed below.

Editor's comments. As you will see, all referees think that the findings are of interest, but they also have several comments, concerns and suggestions, indicating that a major revision of the manuscript is necessary to allow publication in EMBO reports. As the reports are below, and I think all points need to be addressed, I will not detail them here. It will be of particular importance, though, to add the control groups as indicated by referee #3.

Control group

In our original manuscript we presented data from mice with increased expression of the insulin like growth factor-1 receptor (IGF-1R) restricted to the endothelium (hIGFREO), showing that hIGFREO fed a high fat diet do not put on as much weight as their wild type littermates.

We also showed that this leads to advantageous effects on glucose tolerance and insulin sensitivity. Examining the microbiome of hIGFREO fed a high fat high showed a significant difference to their wild type littermates with increased relative abundance of the genus Akkermansia, which has been shown to have similar advantageous effects on glucose tolerance in obese humans and in murine models of obesity.

In response to guidance from the handling editor and reviewer #3 we have now carefully examined hIGFREO and their wild type littermates fed standard chow diet. We show no difference in weight gain, glucose tolerance or differences in the architecture of the microbiota of hIGFREO and wild type littermates (*Summarised in table 1 below*). These data strongly support a significant and important effect of endothelial IGF-1R on the gut response to a high fat diet. We have added this new dataset to the revised manuscript.

Diet	Standard chow Mean (\pm S.E.M)			High fat Mean (\pm S.E.M)		
	hIGFREO	Wild type	P	hIGFREO	Wild type	P
Body mass (g) (Figure 1F)	22.9 (0.49)	21.8 (0.69)	0.21	32.5 (2.05)	40.4 (1.34)	0.007
Fasting blood glucose (mmol/L) (Figure EV1A & 2A)	9.9 (9.38)	9 (0.70)	0.23	4.9 (0.33)	6.9 (0.44)	0.004
Glucose tolerance (AUC) (Figure EV1C & 2C)	53.4 (2.41)	54.9 (3.54)	0.74	42 (1.24)	50.5 (2.96)	0.01
Faiths PD (AU)	8.8 (1.04)	9.1 (0.88)	0.84	9.7 (0.41)	11 (0.34)	0.03

(Figure EV1E & 4B)						
Chao1 (AU) (Figure EV1G & 4D)	162 (29.95)	168 (31.15)	0.89	119.5 (19.56)	182.4 (14.60)	0.03

Table 1. Summary of chow v HFD hIGFREO metabolic data.

We then went on to examine the effect of antibiotic treatment on the microbiome of hIGFREO and wild type littermates fed standard chow. In keeping with the highly insightful comments from reviewer #3, on standard chow, however, hIGFREO and wild type littermates did not tolerate prolonged antibiotic treatment and for welfare reasons had to be culled.

Previous studies of mice on chow treated with antibiotics did not report this problem (*Frontiers in Microbiology*, 8, 2306, 2017) although this group treated Swiss Webster mice as opposed to C57/Bl6 as used in the present report.

We did not add sucrose or sweetener to feed as both of these supplements have been shown to have deleterious effects on glucose tolerance (e.g. *Nature*, 2014; 514:181–186. *Molecular Metabolism* 2019; 27, 22-32).

However, the substantial new data from mice fed chow diet confirm the role of changes in the microbiome occur on a high fat diet and not chow diet in hIGFREO.

Referee #1: *We would like to thank reviewer #1 for their helpful, thorough and insightful review.*

Major concerns:

1. The data from adipose tissues, which show the major difference at the organ level, need to be more carefully examined and interpreted:

In Fig. S2B, authors should quantify the numbers and sizes of adipocytes and present data using histograms to show the distribution, rather than using adipocyte area. According to Fig. S2A, there appear to be more adipocytes of smaller sizes in hIGFREO as compared to the wildtype.

An excellent suggestion. We have now presented adipocyte distribution data in figure EV3B and shown below. We show, as the referee suspected, that hIGFREO have more adipocytes of smaller size (specifically 751-1000 μm^2) than wildtype.

In Fig. S2C, IB4 staining in BAT is apparently different between wildtype and hIGFREO.

Thank you in our revision we have included new images which are better representative of the IB4 data (Figure EV3C).

Furthermore, Fig. S2F-H show trend decrease of leucocyte numbers in the WAT. Scatter plot should be shown to demonstrate the distribution of the data. It is not clear the representative data are based on how many mice examined.

The leukocyte data do show a trend towards decreased populations of leukocytes in hIGFREO mice. However, this is not significant due to the variability of WT mice as you can now see from the individual data points (Figure EVF-H) and shown below.

We have now plotted all results showing individual data points for individual mice as suggested.

It was claimed that "adipose tissue expression of browning markers were not different in hIGFREO compared to WT (Figure 3D)".

However, in Fig. 3D, Pcg1a (possibly misspelled for Pgc1a?) is apparently increased in the WAT. Same markers (Ucp-1, pgc1a, vegfa, adipoq, and lep) should be examined in both WAT and BATs.

We apologise for the typographical error for Pgc1a. We have now measured all genes in both WAT and BAT as suggested, data presented in figure 3D (and below). Presentation of individual data points show the scatter of Pgc1a and no significant difference.

The primary mechanism underlying the anti-obese phenotype in hIGFREO has been attributed to changes in the gut, which should be corroborated. At least, histology should be shown for the intestines from the wildtype and the hIGFREO mice.

Thank you for this excellent suggestion, we have included histology of villi in Figure EV5E-G and shown below, which show there is no difference in villi length or fat content between genotypes.

It is interesting that the antibiotics abolished the observed difference between the wild type and hIGFREO. Data supporting that the EC overexpression of IGFR works through microbiota need to be consolidated.

Thank you for this excellent point. We have now carefully examined hIGFREO and their wild type littermates fed standard chow diet. We show no difference in weight gain, glucose tolerance or differences in the architecture of the microbiota of hIGFREO and wild type littermates (*Summarised in table 1 below*). These data strongly support a significant and important effect of endothelial IGF-1R on the gut response to a high fat diet. We have added this new dataset to the revised manuscript.

Characteristic	Standard chow Mean (\pm S.E.M)			High fat Mean (\pm S.E.M)		
	hIGFREO	Wild type	P	hIGFREO	Wild type	P
Body mass (g) (Figure 1F)	22.9 (0.49)	21.8 (0.69)	0.21	32.5 (2.05)	40.4 (1.34)	0.007
Fasting blood glucose (mmol/L) (Figure EV1A & 2A)	9.9 (9.38)	9 (0.70)	0.23	4.9 (0.33)	6.9 (0.44)	0.004
Glucose tolerance (AUC) (Figure EV1C & 2C)	53.4 (2.41)	54.9 (3.54)	0.74	42 (1.24)	50.5 (2.96)	0.01
Faiths PD (AU) (Figure EV1E & 4B)	8.8 (1.04)	9.1 (0.88)	0.84	9.7 (0.41)	11 (0.34)	0.03
Chao1 (AU) (Figure EV1G & 4D)	162 (29.95)	168 (31.15)	0.89	119.5 (19.56)	182.4 (14.60)	0.03

Table 1. Summary of chow v HFD hIGFREO metabolic data.

In Fig. 5B, WT with or without ABs, and hIGFREO with or without ABs should be plotted on the same graph to enable the demonstration of the effect of ABs per se on body mass.

Thank you we have now plotted the data as suggested (Figure 5B) and shown below.

It seems that WT + ABs already have lower body mass than WT without ABs (in Fig. 1D). This needs to be addressed as premise before further interpretation between WT and h1GFREO, both administered ABs.

Thank you for this excellent attention to detail. As pointed out WT on HFD + ABs have lower body mass than WT on HFD diet without ABs (Figure 5B). However, both WT and h1GFREO on HFD + ABs still have a significant weight increase compared to chow fed mice (Figure 5B) and shown below. We have discussed this in our revised manuscript (Lines 163-169).

In Fig. 6A and B, using heatmap to present the qPCR data is uncommon and fails to reflect the quantitative nature of qPCR and the variations among individual samples. Scatter bar plots should be used instead of heatmaps.

We have now presented all qPCR datasets as scatter plots as suggested and presented in EV5A-C and shown below.

In Fig. 6C, the comparison should be done between hIGFREO vs WT, rather than basal media.

Thank you. The comparison is now between hIGFREO and WT and shown in figure EV5D and below.

To explain how hIGFREO change the intestinal gene expression and the consequent microbiota, data from intestinal ECs needed be provided. Markers for barrier function and immune response should be measured. IGFR should also be determined at the protein level since Tie2 is not entirely specific for ECs.

Thank you for this suggestion, we have now included gene expression data for barrier marker and immune protein in our qPCR data presented in figure EV5A-C and presented below. These data are comparable between genotypes.

We have also measured endothelial protein level of IGF-1R as suggested and this is presented in Figure 1D and also presented below.

The last result section needs clarification. There are 6 experimental groups and the rationale for comparisons is not clear. For example, Fig. 6B "demonstrated an upregulation of Reg3g, Mmp7, Nlrp6 and Pigr". This is an incomplete statement-upregulation in which group? Similarly, "Gene expression of Caco-2 cells treated with conditioned media from hIGFREO endothelial cells showed a significant increase in Reg3g gene expression (Figure 6C), as seen in high fat fed hIGFREO mice (compared to what?).

We apologise for the confusion and have removed the 6 groups. We have now compared Reg3G gene expression between hIGFREO and WT and made this clear in discussion.

The numbers of mice used are not specified for any of the experiments. This should be clearly indicated for each of the experiment. For ANOVA, posthoc test should be specified.

We have now plotted all data showing individual data points for individual mice.

We have updated the statistical methods section to clarify our statistical approach. For plasma concentration-time profile experiments data was analysed using area under the curve and students t-testing. For metabolic parameters measured by indirect calorimetry ANOVA testing was performed using mass as a co-variant (ANCOVA testing) using calrapp.org which is considered the gold standard (Mina *et al.*, 2017. CalR: A Web-based Analysis Tool for Indirect Calorimetry Experiments, bioRxiv).

A justification for only using male mice should be provided.

Thank you, an important point. It is known that estrogen can modulate adiposity in female mice (Stubbins *et al.*, *Eur J Nutr.* 51, 861-70 (2012). Griffin *et al.*, *Am J Physiol Regul Integr Comp Physiol* 311, R211-6 (2016)). However, we performed basic metabolic phenotyping in female WT and hIGFREO mice and found chow fed females had similar body mass but displayed improved glucose tolerance and insulin sensitivity at baseline (Figure 1).

Figure 1. Chow fed female metabolic data.

A, Chow fed female mice with increased expression of the insulin like growth factor-1 receptor (IGF-1R) restricted to the endothelium (hIGFREO) had similar body mass to wildtype littermates (WT). **B**, Chow fed female hIGFREO had similar fasting blood glucose levels as WT. **C & D** Chow fed female hIGFREO had a trend towards improved glucose tolerance, compared to WT but this was not significantly different when evaluated using area under the curve (AUC). **E & F** Chow fed female hIGFREO had a trend towards enhanced insulin sensitivity, compared to WT but this was not significantly different when evaluated using area under the curve (AUC)). N = 6 mice for WT and 8 mice for hIGFREO. Data shown as mean \pm SEM, P<0.05 taken as being statistically significant using student t-test and denoted as *.

hIGFREO females fed with a HFD had a trend towards reduced body mass compared to WT littermates, but this did not reach significance. hIGFREO females on HFD had similar glucose tolerance to WT littermates but displayed improved insulin sensitivity (Figure 2). We have added a line in the revised manuscript regarding the use of male mice only.

Figure 2. High fat diet (HFD) fed female metabolic data.

A, HFD fed female hIGFREO had similar body mass to WT. **B**, HFD fed female hIGFREO had similar fasting blood glucose levels to WT. **C & D** HFD fed female hIGFREO had similar glucose tolerance, compared to WT. **E & F** HFD fed female hIGFREO had a trend towards enhanced insulin sensitivity, compared to WT but this was not significantly different when evaluated using area under the curve (AUC)). N = 6 mice for WT and 8 mice for hIGFREO. Data shown as mean \pm SEM, P<0.05 taken as being statistically significant using student t-test and denoted as *.

Minor points:

1. In Fig. 1D, data should be presented in the same manner as in Fig. 5D, i.e. should measurements in the time course.

Thank you we have now presented both Chow and HFD in figure1D and shown below.

2. In figure 4D, is this difference significant? Description of data showing trends toward difference without statistical significance should be made more accurate, rather than simply stating "no change/no difference".

We apologise for omitting the significance of these data and we have included this in our revised manuscript.

3. Supplemental Table 1 does not seem to be included in the submission. Please provide.

Thank you, this table now included.

4. For Fig. S2C, the color assignment is inconsistent between figure legend and the images. In Fig. S2A-C, same tissues should be assayed, including iWAT, eWAT, and BAT.

Thank you, we have now correctly labelled WAT and BAT.

5. Discussion from Line 249-260 is puzzling. Authors should provide a reasonable conclusion for these findings in the context of the current work.

We apologise for the confusion, in our revised manuscript we have removed this section.

6. "It has been shown that Akkermansia muciniphila can increase Reg3g expression in mice fed a standard chow diet, but not high fat, further suggesting that hGFREO mice have a beneficial phenotype when challenged, which is usually diminished by high fat feeding." This seems contradictory to Fig. 6B showing Reg3g still increased in hGFREO under HFD with ABs. Please clarify.

Thank you. We have now removed this from revised manuscript.

7. Whenever referring to specific data, figures should be cited in discussion.

We have now cited figures in our discussion.

8. Authors should indicate the rationale of using Caco-2 cells.

We used Caco-2 cells as they are heterogeneous mixture of intestinal epithelial cells, a well characterized intestinal *in vitro* model, and have included this in the discussion. Caco-2 cells

are primarily used as a model of the intestinal epithelial barrier. Caco-2 cells were cultured as previously published (Clark *et al.*, *Gut* 2003;52:224-230).

9. Please proofread carefully to avoid any typos and mis-labeling, e.g. Line 171 "paracrine manor (manner?)."

Thank you we have very carefully proofread our revised manuscript.

Referee #2: *We would like to thank reviewer #2 for their helpful, careful and insightful review.*

IGF-1R overexpressed mice have a lower gut microbiota diversity (lower PD) than wild type mice (Figure 4). As a complementary confirmation of their observation that IGF-1R on obesity through gut microbiota, they should consider using some way like fecal transfer of wild type mice to the IGF-1R mice to reverse gut microbiota compositions and then evaluate its effect.

Thank you for this excellent observation. We have used an alternative complementary approach rather than fecal transplant. We now present data from chow fed hIGFREO and wild type littermate mice showing no difference in microbiota diversity (Figure EV1D-G) and shown below.

Applying the antibiotics cocktails depleted all gut microbiota, further decreased microbiota PD diversity, and not only depleted "beneficial bacterial taxa", but also "bad bacterial taxa", which may generate multiple effects on host physiology and may not support their conclusions on the crosstalk. The microbiota sequencing analysis based on the text only shows relative abundances, not the real abundance. So if they claim hIGFREO increases Akkermansia (as potential key player), they should use qPCR to compare total 16S and Akkermansia 16S abundance between wild type and transgenic mice.

Thank you for this important observation. We carried out qPCR analysis, as show below and there was a trend for hIGFREO to have increased Akkermansia, this did not reach significance due to the variability of hIGFREO.

Figure 3. qPCR data for WT v hIGFREO Akkermansia abundance.

We have clarified this in the manuscript and refer to a relative abundance of Akkermansia. This is in accordance with previously published papers on Akkermansia abundance in murine models (Hänninen *et al.*, *Gut* 2018; **67**:1445-1453 and Shin *et al.*, *Gut* 2014;63:727-735).

Referee #3: *We would like to thank reviewer #3 for their helpful and extremely insightful review.*

First and foremost, the experiments are lacking proper control groups. Specifically, WT and IGFR mice on a control diet (and use of Abx on a control diet in the second experiment). These groups are necessary to adequately analyze the effects of the diet and the overexpression.

We have now included data from control chow fed hIGFREO and WT mice in EV1 and shown below. As highlighted as a potential issue by this reviewer, prolonged treatment with antibiotics was not well tolerated by hIGFREO or WT on a chow diet; mice lost weight and had to be sacrificed. Previous studies have used sweeteners to make the cocktail more palatable but this would negate the findings of our current study so we elected not to use this approach. However, the new dataset of chow fed mice support our conclusions.

Figure EV1

The conclusions derived by the authors are not fully supported by the data.

For example, the change in Akkermansia is rather small (a control group would help here), other bacterial species were altered to a much greater extent, so the relative importance of Akkermansia is questioned.

Further, the link to REG3 is tenuous, and the importance/necessity of REG3 is based only on gene expression.

Thank you for this insightful comment which we agree with. It is likely that a single mechanism does not account for the exciting findings we describe. We have therefore moved the Reg3g data to EV5 and offered akkermansia as one possibility, whilst acknowledging further work on other bacterial species is needed.

Introduction: The introduction does not sufficiently explain why the authors hypothesized that IGF1R overexpression would improve metabolic function, or why the microbiota would be involved. Are there existing data that provided a scientific rationale for the study?

An excellent suggestion. We have added a rationale and existing studies of IGF-1R in relation to the microbiome to the introduction as a rationale for this study. Previous studies have shown the IGF-1R has also been shown to modulate the intestinal barrier and conversely the microbiome has been shown to modulate IGF-1R signalling in muscle and bone formation. We therefore wanted to probe this further in the setting of obesity.

Most metabolic outcomes were not different in the IGFR mice, glucose tolerance appeared to be an exception. For that reason, tissue data providing more mechanistic insight into the difference would strengthen the data.

Thank you for another excellent idea. We have shown using western blotting that muscle expression of AKT and phosphorylated AKT is greater in hIGFREO than WT littermates fed the high fat diet. These data are consistent with protection against the adverse metabolic effects of the high fat diet, this is presented in EV2F-G and shown below.

Given the site of overexpression, it would be appropriate to examine markers of vascular/endothelial function.

Thank you for this suggestion, we have now included eNOS and AKT expression from aorta in figure EV2H & I and shown below.

Can the authors explain why pulmonary ECs were isolated? Are these cells phenotypically similar to more commonly examined ECs (e.g. aortic; peripheral microvascular).

Another excellent question. Conditioned media experiments require a large number of EC and pulmonary EC provide an appropriate yield of cells to perform these experiments; this is not feasible using cultured murine small intestinal EC. We have added a comment in methods explaining this.

Whilst there is heterogeneity across vascular beds, pulmonary EC are transcriptionally similar to EC from many other organs as shown by publicly available data from Shahin Rafii's group (EndoDB (<https://pubmed.ncbi.nlm.nih.gov/30357379/>)).

Regarding antibiotic treatment, how were they administered? Also, a universal primer would confirm the success of the antibiotic treatment. Did the mice tolerate the Abx treatment well? Several studies have reported that mice do not tolerate Abx treatment, and sugar must be added to encourage feeding. The magnitude of weight gain in the Abx experiment looks smaller than the first experiment, which could support this notion (again a control group would help here).

Thank you for this. The antibiotics were administered in drinking water and this is now included in the method and results section. On HFD mice tolerated the antibiotics well and confirmation of successful treatment is shown in Figure 4 below.

Figure 4. qPCR data from WT and hIGFREO mice on HFD and HFD and antibiotic treatment confirming antibiotics treatment was successful.

As suspected by reviewer #3 on standard chow hIGFREO and their wild type littermates did not tolerate prolonged antibiotic treatment and for welfare reasons had to be culled. We did

not add sugar to encourage feeding as this would significantly confuse the findings. However, our chow control group dataset provide supportive data (EV1).

Can the authors explain the significance of the lipid absorption test? And why 3 hours? I would have thought TG are still being produced at 3h, and many postprandial tests measure out 4 to 6 hours.

Thank you for this excellent comment. We found that at 4 hours obtaining blood from mice was not well tolerated and bruising occurred and therefore for welfare issues we terminated the test at 3 hours. Nonetheless the lipid absorption test demonstrates a physiologically important difference between HIGFREO and wildtype in lipid absorption consistent with the overall phenotype of HIGFREO.

Dear Prof. Kearney,

Thank you for the submission of your revised manuscript to our editorial offices. We have now received the reports from the three referees that were asked to re-evaluate your study, you will find below. As you will see, the referees now support the publication of your study in EMBO reports. Referee #1 has some further suggestions to improve the manuscript we ask you to address in a final revised version.

Moreover, I have these editorial requests I ask you to also address:

- Please provide a more comprehensive and shorter title with not more than 100 characters (including spaces).
- Please provide the abstract written in present tense.
- Please move all the funding information to the acknowledgements (and remove the section 'grant support' from the title page). Please also make sure that equivalent and complete funding information is provided in our submission system.
- Please remove the section 'Disclosures' and provide the information in two separate paragraphs. We require separate 'Conflict of interest' and 'Data availability' sections.
- We would like to publish your manuscript as Report. For a Scientific Report we require that results and discussion sections are combined in a single chapter called "Results & Discussion". Please do this for your manuscript. For more details, please refer to our guide to authors:
<http://www.embopress.org/page/journal/14693178/authorguide#researcharticleguide>
- Please move all the methods information from the Appendix to the main manuscript. The Appendix file can then be deleted.
- Please add scale bars also to Figs. 1G and EV5E. Moreover, in the right panel of EV5E the lower right corner seems cut off. Please check.
- Please call out the EV figures and their panels sequentially or arrange these differently in the figures. In particular, provide separate callouts for the panels of Fig. EV3.
- There is no call-out for Table EV1. Or there is one for 'Supplementary Table 1'. Please check.
- Some microscopy images could have higher image quality. Please try to provide the highest resolution possible.
- Please add the headings 'Figure Legends' and 'Expanded View Figure Legends' to the respective figure legends.
- Please also note our new reference format. Please format your reference list accordingly:
<http://www.embopress.org/page/journal/14693178/authorguide#referencesformat>
- As the few Western blots shown are significantly cropped, could you please provide the source

data for the blots (in main and EV figures). The source data will be published in a separate source data file online along with the accepted manuscript and will be linked to the relevant figure. Please submit the source data (scans of entire blots) together with the final revised manuscript. Please include size markers for the scans of entire blots, label the scans with figure and panel number, and send one PDF file per figure.

- Please make sure that regarding data quantification and statistics, where applicable, the number "n" for how many independent experiments were performed, the nature of the replicate (biological or technical), the bars and error bars (e.g. SEM, SD) and the test used to calculate p-values is defined in the respective figure legends.

- Please also mark those conditions where after statistical testing the differences were not significant (e.g. with 'ns').

- For at least one condition for EV4C it seems only two data points are shown. In such a (n=2), statistical testing or showing error bars does not make much sense (with two replicates). In that case, please show these data without statistics, by showing the two datasets separated. This is much more transparent and illustrates better the data. Or add a third replicate to do proper statistics.

- Finally, please find attached a word file of the manuscript text (provided by our publisher) with changes we ask you to include in your final manuscript text, and some queries, we ask you to address. Please provide your final manuscript file with track changes, in order that we can see any modifications done.

In addition, I would need from you:

- a short, two-sentence summary of the manuscript
- two to three bullet points highlighting the key findings of your study
- a schematic summary figure (in jpeg or tiff format with the exact width of 550 pixels and a height of not more than 400 pixels) that can be used as a visual synopsis on our website.

Kind regards,

Achim Breiling
Editor
EMBO Reports

Referee #1:

The manuscript has been significantly improved after the revision. Some minor comments should be addressed to further enhance the completeness and clarity of the manuscript:

1. In the introduction, "The endothelium, previously thought to be an inert monolayer, has emerged as a complex paracrine/autocrine organ, important in the regulation of a range of homeostatic processes." In the context of obesity and diabetes, this can be supported by a recent report demonstrating an active role of endothelial control of metabolism (Tang et al . Circulation.

2020;142:365-379) in addition to those cited. Also, some background knowledge related to the gut microbiota in relation to the current study should be briefly discussed to enhance the readability to audience unfamiliar with microbiota.

2. Abbreviation should be defined at first appearance to enable an easier understanding of the manuscript. E.g. Reg3g was not defined until the last paragraph in the discussion. Better move that to the results.

3. The response regarding the control groups of normal diet with antibiotics should be included in the discussion, in addition to point-by-point letter to reviewers and editors. This is an important issue and could benefit the readers if clarified in the manuscript.

4. Figure EV2, F and G, the blot for b-actin should be shown.

5. In Figure EV3D, the IB4 was green and Lipid Tox was red, which seems to be reversely labeled.

6. It is best to indicate the number of animals in each group in the figure legend, which has become a standard for high-impact journals.

7. Figure EV5 B and C are difficult to read, with most of the data points compressed at the bottom of the y axis. This can be improved by adjusting the y axis scale.

8. Distinguishing chow vs high-fat diet using different colors may enhance readability of the figures.

Referee #2:

The revised version has fully addressed previous comments and concerns. I will recommend it for publishing.

Referee #3:

The authors have made a tremendous effort to address the reviewers' questions and comments, including new experiments outcome variables, and have satisfied this reviewer's concerns.

The authors have addressed all minor editorial requests and the remaining minor points by Referee #1.

Prof. Mark Kearney
The University of Leeds
Multidisciplinary Cardiovascular Research Centre
LIGHT laboratories
Clarendon Way
Leeds LS2 9JT
United Kingdom

Dear Prof. Kearney,

I am very pleased to accept your manuscript for publication in the next available issue of EMBO reports. Thank you for your contribution to our journal.

At the end of this email I include important information about how to proceed. Please ensure that you take the time to read the information and complete and return the necessary forms to allow us to publish your manuscript as quickly as possible.

As part of the EMBO publication's Transparent Editorial Process, EMBO reports publishes online a Review Process File to accompany accepted manuscripts. As you are aware, this File will be published in conjunction with your paper and will include the referee reports, your point-by-point response and all pertinent correspondence relating to the manuscript.

If you do NOT want this File to be published, please inform the editorial office within 2 days, if you have not done so already, otherwise the File will be published by default [contact: emboreports@embo.org]. If you do opt out, the Review Process File link will point to the following statement: "No Review Process File is available with this article, as the authors have chosen not to make the review process public in this case."

Should you be planning a Press Release on your article, please get in contact with emboreports@wiley.com as early as possible, in order to coordinate publication and release dates.

Thank you again for your contribution to EMBO reports and congratulations on a successful publication. Please consider us again in the future for your most exciting work.

Yours sincerely,

Achim Breiling
Editor
EMBO Reports

THINGS TO DO NOW:

You will receive proofs by e-mail approximately 2-3 weeks after all relevant files have been sent to our Production Office; you should return your corrections within 2 days of receiving the proofs.

Please inform us if there is likely to be any difficulty in reaching you at the above address at that time. Failure to meet our deadlines may result in a delay of publication, or publication without your corrections.

All further communications concerning your paper should quote reference number EMBOR-2020-50767V3 and be addressed to emboreports@wiley.com.

Should you be planning a Press Release on your article, please get in contact with emboreports@wiley.com as early as possible, in order to coordinate publication and release dates.

Corresponding Author Name: Prof Mark Kearney

Journal Submitted to: EMBO reports

Manuscript Number: EMBOR-2020-50767V1